# Color Measurement and Analysis of Fruit with a Battery-Less NFC Sensor

**DOI:** 10.3390/s19071741

**Published:** 2019-04-11

**Authors:** Antonio Lazaro, Marti Boada, Ramon Villarino, David Girbau

**Affiliations:** Department of Electronic, Electric and Automatic Control Engineering, Universitat Rovira i Virgili, 43007 Tarragona, Spain; marti.boada@urv.cat (M.B.); ramon.villarino@urv.cat (R.V.); david.girbau@urv.cat (D.G.)

**Keywords:** battery-less, color sensor, Near Field Communication, Radio Frequency Identification (RFID), energy harvesting, food quality, classification, Support Vector Machine (SVM), machine learning, Internet of Things (IoT)

## Abstract

This paper presents a color-based classification system for grading the ripeness of fruit using a battery-less Near Field Communication (NFC) tag. The tag consists of a color sensor connected to a low-power microcontroller that is connected to an NFC chip. The tag is powered by the energy harvested from the magnetic field generated by a commercial smartphone used as a reader. The raw RGB color data measured by the colorimeter is converted to HSV (hue, saturation, value) color space. The hue angle and saturation are used as features for classification. Different classification algorithms are compared for classifying the ripeness of different fruits in order to show the robustness of the system. The low cost of NFC chips means that tags with sensing capability can be manufactured economically. In addition, nowadays, most commercial smartphones have NFC capability and thus a specific reader is not necessary. The measurement of different samples obtained on different days is used to train the classification algorithms. The results of training the classifiers have been saved to the cloud. A mobile application has been developed for the prediction based on a table-based method, where the boundary decision is downloaded from a cloud service for each product. High accuracy, between 80 and 93%, is obtained depending on the kind of fruit and the algorithm used.

## 1. Introduction

The surface color of food is the first element that the consumer observes and has a great influence on the consumer’s selection [1]. It depends on various factors, including the temperature, humidity, and biochemical changes that occur during growth, maturation, and postharvest handling and processing [1]. In consequence, food color is an excellent indicator of its quality because it takes into account these parameters and it is one of the most broadly measured product quality attributes in postharvest handling and in food processing research and industry [1]. 

In recent years, the consumers’ interest in ecological food, the origin, and the traceability have grown. Radio Frequency Identification (RFID) technology could play an important role in the following years. With the long marketing chain for many fruits and vegetables that is currently operating, the use-by-date or recommended date of consumption is difficult to establish for these products. Surface color can be a good parameter to give an indication of the freshness and flavor quality of fresh products. For example, the ripening process of tomatoes is clearly characterized by the color evolution of the fruit surface. After harvesting, the fruit continues to ripen and its color turns from green to red. Because of transportation time, some matured vegetables and fruits (for example red tomatoes) are sold to local supermarkets and green fruits can be shipped to shoppers at higher distances [2]. Color has been used to evaluate fruit maturity for tomatoes [3], bananas [4], apples [5], pears [6], oranges [7], and dates [8]. Some colors are associated with food products. For instance, yellow is associated with ripe bananas in a good state and good tomatoes are associated with red instead of orange. 

Food color measurements have been made using a large variety of instruments [1]—colorimeters [9], spectrophotometers [10,11] and color measurement by computer vision with digital cameras [3,4,5,6,7,8,12]. However, sometimes these instruments are expensive or the measurement setups are not suitable for performing real-time measurements by a consumer in a supermarket or at home.

RFID (Radio Frequency Identification) technology [13] is a well-known wireless application for traceability, logistics, and access control. It is recognized as a key technology for the development of the Internet of Things (IoT). In recent years, RFID technology has spread greatly. Traditional barcodes are progressively being replaced by low-cost RFID tags to track items. Food traceability systems are currently supported by governments, consumers, and producers [14]. There are different types of RFID technology (Low Frequency, High Frequency or Ultra High Frequency according to its frequency band); one which has expanded greatly is the Near Field Communication (NFC) technology. NFC is a short-range Radio Frequency Identification system (RFID) that allows communication between devices using the industrial, scientific and medical (ISM ) 13.56 MHz RFID band [13]. Although the first NFC devices appeared on the market more than a decade ago [15,16], they have not reached maturity until the massive incorporation for payment systems. Today, NFC technology allows safe fast data transfer between devices. A simple tap allows the consumers to fulfill contactless transactions and access to digital content without the need of paring the devices. As a consequence, the presence of the NFC technology is growing in the Internet of Things (IoT) framework and Industry 4.0 [17,18,19]. For this reason, smartphones incorporate an NFC reader [20]. This fact has aroused the interest of NFC technology for the sensor market. NFC tags can integrate low-power and low-cost sensors that can be read placing the smartphone or reader close to the tag, without the devices needing to be paired. The data can be saved within the standard NFC message in NDEF (NFC Data Exchange Format) format and can be read and processed by user-friendly mobile applications. Then, the data can be stored in cloud services opening a large spectrum of new IoT applications. In addition, the most important NFC integrated circuit (IC) manufacturers have commercialized NFC IC with an energy harvesting capability that can provide energy to small sensors and microcontrollers [20].

Colorimeters and spectrophotometers are devices that can be used to measure the color of a test sample. A spectrophotometer is an instrument with high precision and is adequate for more complex color analyses because it can measure the spectral reflectance at each wavelength. However, spectrophotometers are more expensive than colorimeters. Therefore, a colorimeter is the best choice for quality inspection. In a recent previous work, the authors have presented a battery-less NFC tag that integrates a colorimeter for PH measurement [21]. Based on this colorimeter another application is presented here. In this work, a low-cost battery-less NFC tag is designed for food color measurement and classification based on the measured color and using a smartphone. A simple tap is required for the color measurement using the smartphone equipped with NFC to read the color measured by a colorimeter integrated into the tag. This low-cost device can be used to classify different fresh products in the supermarket or at home. Section 2 summarizes the design of the tag. Experimental results of different fresh products are presented in Section 3. In order to obtain a robust method that can be implemented easily in a mobile telephone, different classification methods are compared in Section 3. Finally, the summary of the work and conclusions are provided in Section 4.

## 2. System Overview

A system overview is shown in Figure 1. The food being tested is placed over the aperture of the colorimeter integrated into the NFC tag. Its color is a function of the temperature, humidity, and days since it was harvested. When the user taps his smartphone equipped with an NFC reader, then the NFC IC in the tag takes energy from the RF interrogating signal that is used to feed the circuitry and returns the color measurement as an NFC message. The application in the smartphone reads the message, classifies the quality of the food, and gives additional information about the origin or some other useful information. This information, as well as the parameters needed for food classification, can be automatically downloaded from the cloud. 

In this work, an NFC tag previously presented in Reference [21] has been modified and evolved. A schema of the tag is shown in Figure 2. The tag designed consists of a color light-to-digital converter TCS34725 (TAOS Inc., Plano, TX, USA) from TAOS [22], a white 4150 K LED used to illuminate the sample, a low-power microcontroller Atmel Tiny 85 (Atmel Corporation, San Jose, CA, USA), and an NFC chip from ST (M24LR04E-R, STMicroelectronics, Geneva, Switzerland). The TCS34725 has RGB and clear light sensing elements (Figure 2). The light-to-digital converter is performed using a 3 × 4 photodiode array. This array consists of red-filtered, green-filtered, blue-filtered, and clear (unfiltered) photodiodes. To improve the accuracy in the color measurement, the TCS34725 is coated with an infrared (IR) radiation blocking filter that reduces the IR spectral components of the incoming light. Figure 2b shows the spectral response curves of the four channels of the sensor from the datasheet [22]. The amplified photodiode currents are simultaneously converted with a 16-bit analog-to-digital converter (ADC). These three chips are interconnected through an I2C bus. The NFC IC is connected to a PCB loop antenna, which works at 13.56 MHz, corresponding to the NFC frequency standard. 

A photograph of the prototype is shown in Figure 3. A protection envelope has been designed using a 3D printer. This case can be customized and has a window over the color sensor that is covered by the fruit. The box is opaque and it is only illuminated by the internal white led. This fact improves the accuracy and repeatability of the measurements. 

In the prototype antenna, a square loop of 50 mm × 50 mm printed on FR4 PCB was designed with the Keysight Momentum electromagnetic simulator. It consists of 6 loops with individual widths of 0.6 mm. The tag’s antenna inductance (*L_a_*) is measured with a Vector Network Analyzer (VNA) and its value is 2.9 µH, in agreement with the simulations. A tuning capacitance (*C_tun_*) of 15 pF is added to the internal capacitance of the IC (*C_IC_* = 27.5 pF for M24LR04E-R) to adjust the resonance frequency (*f_r_*) at 13.56 MHz according to the expression:
(1)$$f_{r}\approx \frac{1}{2\pi \sqrt{L_{a}\left(C_{IC}+C_{p}+C_{tuning}\right)}}$$
where *C_p_* considers the layout parasitic capacitance, which includes the antenna capacitance and the parasitic capacitance due to the interconnections between the antenna and the NFC IC. Finally, the resonance frequency is checked with a Vector Network Analyzer (VNA), measuring the S_11_ parameter by using another loop antenna approached to the tag.

Due to the limitation on the power that the NFC IC can harvest from the RF signal, a low-power microcontroller must be used to receive the data from the colorimeter IC and save the data in the internal memory of the NFC IC. In the prototype designed in this work, the Atmel 8-bit AVR ATtiny85 microcontroller is chosen. Among its features, it stands out that this microcontroller can work down to 1.8 V at 1 MHz. The clock speed has been chosen at 1 MHz to reduce the current consumption to approximately 300 μA at 3.3 V. The current consumption of each component is shown in Table 1. The total current consumption of the tag is around 3 mA, which is under the 5 mA that the NFC IC can harvest from the RF. Part of this consumption is due to the white led used to illuminate the sample. 

For this power consumption, a read range up to 2 cm is obtained depending on the smartphone used as a reader. For example, Figure 4a shows the measured output voltage of the NFC IC harvesting output. It can be observed that it stays nearly constant at 3 V until the average magnetic field reaches the minim value (*H_min_)*. Below this value, the voltage at the input of NFC IC is not enough for the right RF to DC conversion and the harvesting output quickly vanishes. The average magnetic field (*H_av_*) can be measured with an antenna with a known antenna factor. In our case, the same antenna used in the tag is used as a test antenna. The tag is replaced by the test antenna and the input antenna impedance (*Z_in_*) is measured with a VNA. The antenna factor (AF) is calculated using [20]:
(2)$$AF=\frac{Z_{0}+Z_{in}}{j2\pi f\mu_{0}Z_{0}A\cdot N}$$
where *Z*_0_ is the reference impedance (50 Ω), *f* is the frequency, *A* is the loop area, *N* is the number of turns of the loop and *µ*_0_ is the vacuum magnetic permeability.

After the antenna is known for each distance, the average magnetic field is obtained from the Root-Mean-Square voltage (*V_RMS_*) calculated from the measured power at the carrier frequency with a Spectrum Analyzer connected to the test antenna.
(3)$$H_{av}\left(A_{RMS}/m\right)=V_{RMS}\cdot \left|AF\right|$$


Figure 4b shows that the antenna factor changes with the distance to the mobile due to the presence of metal (a smartphone with metallic case is used). The current induced in the metal reduces the effective magnetic flux and therefore the inductance and the antenna factor decrease when the antenna is closer to the metal. The average magnetic field is shown in Figure 4c. This read range corresponds to a measured minimum magnetic field of 1.1 A/m (Figure 4a) [20,21]. The bandwidth is determined by the NFC standard (standard ISO15693 in this case). 

Theoretically, as the data rate between sensor and reader is not excessively high, the bandwidth could be reduced. Consequently, coils with high-quality factors to increase energy transfer efficiency can be designed. However, the loaded quality factor of the tag is low [20,21] and it is determined by the low IC equivalent resistance. Nevertheless, this low-Q factor has an advantage because the system is more robust to the detuning due to the proximity of metals (metallic case of the modern smartphones) or high permittivity materials (for example the body).

The battery-less system presented is based on low-cost commercial integrated circuits. Due to the wide diffusion of NFC systems, the cost of NFC IC and the low-power microcontroller are smaller than 1 $. The price of the colorimeter IC is around 1.5 $. Thus the overall cost of the tag including the envelope can be under 5 $ considering large volumes of production (see estimation in Table 1). This cost is noticeably lower than professional colorimeters or spectrometers that are typically starting from 600–1000 $. In addition, the presented system is easy to use and is highly customizable depending on the final application. The lack of battery is another advantage because it avoids the need to replace or recharge the batteries whereas, enlarging the durability of the devices and avoiding the component with higher cost. In addition, the batteries contain toxic components that can contaminate the food in addition to generate non-recyclable waste.

## 3. Experimental Results

In order to test the system, different typical fruits (bananas, and red and golden apples) were measured with the NFC colorimeter. Several samples on different days at room temperature and within the fridge were tested. The objective was to classify the quality of the pieces of fruit depending on the number of days outside the fridge and using the color information. A fruit is considered to belong to the good class if the days at room temperature outside the fridge are equal to or less than six. The aim is to find a simple but accurate method that, after a training process performed by the product manufacturer, the consumer can apply using a smartphone without sophisticated tools like Matlab. The calibration was done by a different colorimeter (same model but a different one) but connected directly to the computer, through the programing connector. The data is transferred to the database to perform the training of the classifiers. The samples are taken in different positions to consider the variations of color on the surface. It is important to note that the samples have been kept considering typical conditions (typical range of temperatures and humidity) that the end user will find. To ensure these steps it is preferable that the calibration is carried out by the manufacturer in a qualified laboratory.

The color of an object can be described by several color coordinate systems (called color spaces). Figure 5 shows the representation of the RGB, HSV, and CIELab color spaces. The first decision is to choose the color space. The most popular is RGB (red, green, and blue), which is used in video monitors. The raw RGB information given by the colorimeter is rarely used in the literature on food classification. HSV [23] and Lab [24] are alternative representations of the RGB color model designed to be more closely aligned to the way human vision perceives color-making attributes. HSV and Lab color spaces are the most used in the literature for this application [25,26,27]. The most frequently used is the CIELab color space, due to its uniform color distribution and because its color perception is closest to that of the human eye. The color in the CIELab color space is determined by three values: L* is the lightness, and a* and b* are the green-red and blue-yellow color components, respectively. Humans identify a colored object through its chromaticity and brightness [3]. The chromaticity can be further divided into two parts: Hue and saturation [3]. HSV is a cylindrical representation where the angle represents the hue, starting at 0° for red, the height is the value and the saturation is given by the distance to the cylinder axis (see Figure 5b).

The ratio a/b was used as a color index in apples, tomatoes, citrus, and carambola fruit [1]. A high correlation between peel color related to maturity and Hunter’s a/b ratio for mangos has been found in Reference [28]. However, this ratio is a function of the hue, which is the angle that can be computed from ratio *atan*(a/b). Therefore, we decided to use the HSV model in this work.

Colorimeters have been used in other applications with good accuracy compared to other color measurement systems such as spectrometers [29,30]. However, it is needed to check the repeatability and accuracy of the measurements. It has been investigated through the measurement of three samples with different colors. Each sample has been measured 200 times with three colorimeters using the same IC model. Table 2 summarizes the main results. This table shows the average values of each HSV sample, the normalized standard deviation with respect to the maximum range of the parameter, and the normalized maximum difference between the colorimeters. It can be observed that the differences and the deviation is smaller than the typical difference observed with the days. The error is uniformly distributed, showing that the main source of error is discretization noise due to the internal analog to digital conversion in the colorimeter IC. The difference in the HSV values between different ICs are small (typically under 1%). It is assumed that the calibration and the measurement by the final user is done with the same model of colorimeter to improve the repeatability. However, small differences can be found between colorimeters and spectrometers [30].

Figure 6 shows the histograms of HSV measurements of a golden apple as a function of the days and the ripening conditions (in the fridge and at room temperature, respectively). Three kinds of fruit are considered (bananas, red apples, and golden apples). The measurements were taken with the colorimeter of the NFC tag. In Figure 7 the Cumulative Distribution Function (CDF) of the hue and saturation parameters for the golden apple in the fridge and at room temperature as a function of the number of days is shown. It can be observed that the color variation is smoother for the fruit conserved in the fridge than that at room temperature. Therefore, the ripeness grade is also a function of the environment parameters. Figure 8, Figure 9, Figure 10 and Figure 11 show the histograms of HSV measurements and CDF of the hue parameter for the banana (Figure 8 and Figure 9) and the red apple (Figure 10 and Figure 11). Similar behavior is obtained for the red apple case. In the case of banana, the fruit is degraded very fast for more than nine days at room temperature. This fact can be appreciated better in the variation of the saturation value for the banana at room temperature. As a conclusion, the combination of hue and saturation value can be used to study the degree of ripeness.

HSV and CIELab values were studied in order to select the most suitable values to analyze the ripeness state of the fruit. Table 3 shows the average of 100 samples for each fruit stored at room temperature, comparing the measurements of the first day (day 0) against the measurements after a few days (day 15 for apples, and day 9 for bananas). The Δ% row shows the percentage increment of each parameter, taking into account the range of each parameter. It can be observed that the best combination is obtained using the H and S values. Hue changes 1.6% for red apples, 5.4% for golden apples and 8.7% for bananas; whereas saturation (S) changes 2% for red apples, 12% for golden apples and 42% for bananas. However, the variations are smaller using CIELab color space, with differences between 0.1% and 2% for L, 1.2% and 4.3% for parameter a* and 1.4% and 8.2 for parameter b*. 

Extracting and selecting features helps to improve a machine learning algorithm by focusing on the data that are most likely to produce accurate results. The challenge is to find the minimum number of features that will capture the essential patterns in the data. From the histograms presented above, it can be concluded that the hue angle and saturation are good choices. The variation over time is higher compared to the value parameter. Another possibility is to apply Principal Component Analysis (PCA) analysis to find the principal components in order to reduce the number of features. However, PCA is not considered to simplify the implementation in a mobile environment because it can increase the computational complexity. The next step is to select a machine learning algorithm. No single machine learning algorithm works for every problem; therefore, the best algorithm is found by exploring the datasets for different algorithms. The comparisons were performed using the statistics and machine learning toolbox in the Matlab R2017a software. The training is performed using a computer with Matlab and ideally should be performed by the manufacturer or a laboratory; however, once the training is finished, the parameters are sent to the cloud server, from where the consumer can download the parameters that are necessary for predicting the class. Therefore, as the prediction step will be made on a mobile, the simplicity of the algorithm and minimal tuning play an important role. Another possibility is that a cloud server executes a program that returns the results as a function of the color measurement. However, mobile computing is preferred. Two classes were considered: Good state (days < 6) and bad state (days >= 6) assuming that the fruit is outside the refrigerator. Other intermediated classes can be defined if necessary. Half of the measurements in the dataset were used for training and the other half were used for testing the prediction. In this work the following algorithms are considered—linear SVM [31], linear discriminate analysis (LDA) [32], Naive Bayes [33], decision tree [34] and Nearest neighbor [35]. Neuronal networks are not included because they often require a lot of tuning parameters. 

Although SVM and LDA have the same boundary decision (a line), their assumptions are different. LDA assumes that data are distributed normally. SVM is a powerful algorithm that does not introduce any assumptions into the data [31]. The LDA algorithm is somewhat sensitive to outliers because it uses all the data in the set to estimate the covariance matrices [32]. On the contrary, SVM uses a data subset that includes those data points that lie on the separating margin. The data points used for optimization are called support vectors because they determine how the SVM discriminate between groups and thus support the classification. 

The Naive Bayes classifier is based on Bayes’ theorem assuming that the predictors are independent random variables within each class. Even if this assumption is not followed, it usually works well [33] and it is widely used. 

The Decision Tree Classifier is a simple and widely used classification technique. The Decision Tree Classifier repetitively divides the working area (plot) into subparts by identifying lines. In general, decision trees are constructed via an algorithmic approach that identifies ways to split a data set based on different conditions. It is one of the most widely used and practical methods for supervised learning. Other types of non-parametric supervised learning methods are the decision trees [34] that are often used for classification and regression problems. The objective is to obtain a model based on simple decision rules based on the characteristics of the data that predict the value of an objective variable. Decision trees divide the feature space into axis-parallel rectangles or hyperplanes.

The nearest neighbor search locates the k-nearest neighbors or all neighbors within a specified distance from certain query data points, based on the specified distance metric. A point is assigned to the most common class in the k nearest neighbors, where k is a small positive integer [35]. The simple case is for k=1, where the point is assigned to the class of the nearest neighbor point. One of the simplest decision procedures that can be used for classification is the nearest neighbor (NN) rule. This classifies a sample based on the category of its nearest neighbor.

In order to avoid introducing any bias in the classification results, the dataset is composed of the same number of measurements each day (100 measurements). These samples have been obtained in random positions of the surface of the fruits to take into account their heterogeneous composition. In the preliminary results, 12 fruits per day are measured. Another fruit that is not included in the training dataset is used for testing. Figure 12, Figure 13, Figure 14, Figure 15, Figure 16 and Figure 17 show the decision regions found for each algorithm superposed on the scatter plot of the training measurements of the dataset.

To explore the tradeoff between different kinds of misclassification, a confusion matrix is used. Table 4 summarizes the confusion matrix calculated with different methods—linear SVM, Naive-Bayes, decision tree, and Nearest Neighbor. In order to compare the methods, the accuracy is also included. The accuracy of the confusion matrix is computed using (4) and the results of accuracy for each feature are illustrated in the last column in Table 4.
(4)$$Accuracy\left(\%\right)=100\frac{TP+TN}{TOTAL\;DATA}$$


The highest accuracy is obtained for the banana and the worst for the red apple. In the case of the golden apple, the two features (hue and saturation) are clearly correlated because all the training points fall along a line. Therefore, a single feature (for example the hue or using PCA decomposition) can be used for classification. A smooth variation of the color is found as a function of the time. The decision boundary found for all classifiers is a perpendicular line to this line. Accuracies of about 82–84% are obtained depending on the classifier algorithm considered. In the case of the banana, as it can be observed from the histogram, the color remains nearly constant for the first days, and then starts to degrade fast from the sixth day. The change in the color is in both hue and saturation features and the relation is not so linear as in the case of the golden apple. However, the two classes are clearly separable and high accuracy is obtained for all the classifiers. The last case is more complicated to classify because the color is not homogeneous. This fruit has regions of red but other regions are close to yellow. To avoid this problem only the red region is considered and the points are filtered before the training in the range of hue < 50°. If a point falls outside this range, the application asks the user to repeat the measurement in another point in the red region. After outliers have been filtered, the color degradation is smooth as in the case of the golden apple. However, due to the heterogeneous distribution of the color on the surface, the accuracy obtained for the different classifier algorithms is considerably lower than in the previous cases. Only LDA and Nearest Neighbor with k = 5 obtained reasonable values of 73 and 71%, respectively. An improvement is found if a quadratic discriminant analysis (QDA) is used, achieving an accuracy of 80% because the boundary decision can be described better for a parabolic function.

Machine learning applications can be deployed in production systems on desktops, in enterprise IT systems (either onsite or in the cloud), and embedded systems. For prediction, the linear SVM, LDA or QDA give an analytic boundary decision function (a line or quadratic function) that only needs to know the polynomial coefficients to define the decision region. The evaluation of the algorithm in the other methods requires the training data from the server. In addition, it requires the implementation in JAVA, C, PHP or another language in the mobile environment or in the cloud. In order to avoid this problem, a table (or image) can be downloaded from the cloud service that is used to interpolate the results. 

An Android application has been developed to analyze the sample and show the result to the user. The operation flow is described in the flowchart of Figure 18. When the application is launched it connects to a server to obtain a list of available fruits and downloads an image and some information about each of them. Then it shows the list of the possible fruits to analyze (Figure 19a), the user selects the fruit and a message appears to place the smartphone over the tag (Figure 19b). When the tag is triggered, it performs the color reading, and sends the Near Field Communication Data Exchange Format (NDEF) message with the HSV and RGB values to the smartphone, which processes the message and shows the result on the screen (Figure 19c). The hue and saturation values are calculated to extrapolate the point using the image previously downloaded, which represents the decision boundaries of the training (Figure 19d). When the user pushes over the image of the fruit a new screen is open (Figure 19e), where he can consult additional information about the product such as the origin, date of collection, web links, etc. The application obtains the resolution and the minimum and maximum values of hue and saturation for each image. This makes it possible to check the color of a specific point of the classification matrix and determine the class the measured color belongs to. The application sends a warning to the user when the point is outside the decision boundaries, because the analyzed point falls on some area with high concentration of pigments or there is a defect on the surface with different color. After that, the user can select another point (see the flowchart in Figure 18). This method is computationally cheap on the reader side. Furthermore, the fact that the axis boundaries and the image resolution are retrieved from the server makes it possible to adapt this system to any kind of image, no matter which classification algorithm has been used or how many classes it contains, whereas the colors of each class are well differentiated.

Figure 20 shows a smartphone with the main screen of the application after measuring the color of a red apple that is on top of the 3D printed enclosure that contains the tag. 

## 4. Conclusions

In this work, a system for classifying the fruit ripeness grade based on the color measured with a battery-less NFC tag and read from a mobile phone is presented. The tag integrates a microcontroller, a colorimeter, a led, and an NFC IC. The tag is powered by the energy harvested from the mobile. Experimental results show that the ripeness grade is a function of time and environment conditions (especially the storage temperature). This work uses HSV color space for classification. It is observed that the main parameters that change are the hue and saturation, which are used as features for the classification. Different classification algorithms have been compared in order to show the robustness of the system. Linear discriminant analysis and nearest neighbor work well in all cases. The proposed system is a low-cost solution compared with expensive spectrometers. In addition, the measurement is not influenced by external illumination, and therefore the measurements are repeatable in comparison with computer vision systems based on mobile cameras. A simple table-based method is proposed to avoid increasing the complexity of implementing the software in the mobile application when the boundary decision regions are not described by analytical functions, such as the case of the nearest neighbor classifier. A drawback of the presented colorimeter is the small size of the analyzed area. If large defects or pigments with different color fall in this area, the points are treated as outliers and the measurement must be repeated. The portable device and algorithm described in this work could be extended to another applications, such as obtaining the grade of ripeness of the fruit during the harvesting, whenever the classification algorithms were trained with samples of a different grade of maturation. This work shows the potential of sensors based on NFC technology for novel IoT applications. 

## Figures and Tables

**Figure 1 sensors-19-01741-f001:**
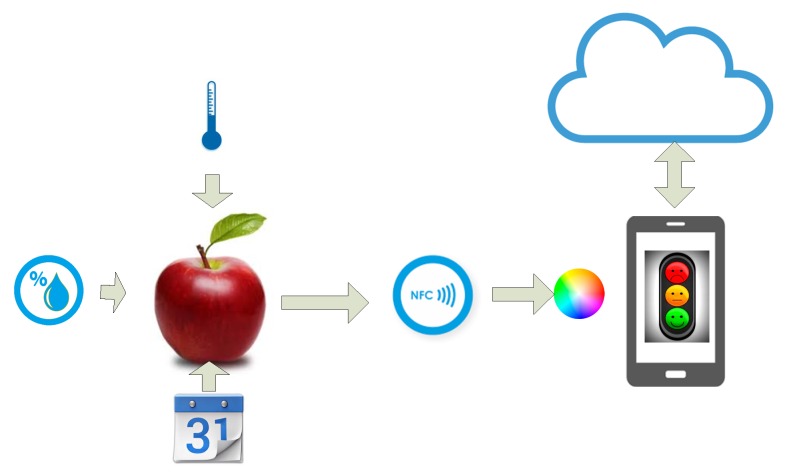
Block diagram of the system.

**Figure 2 sensors-19-01741-f002:**
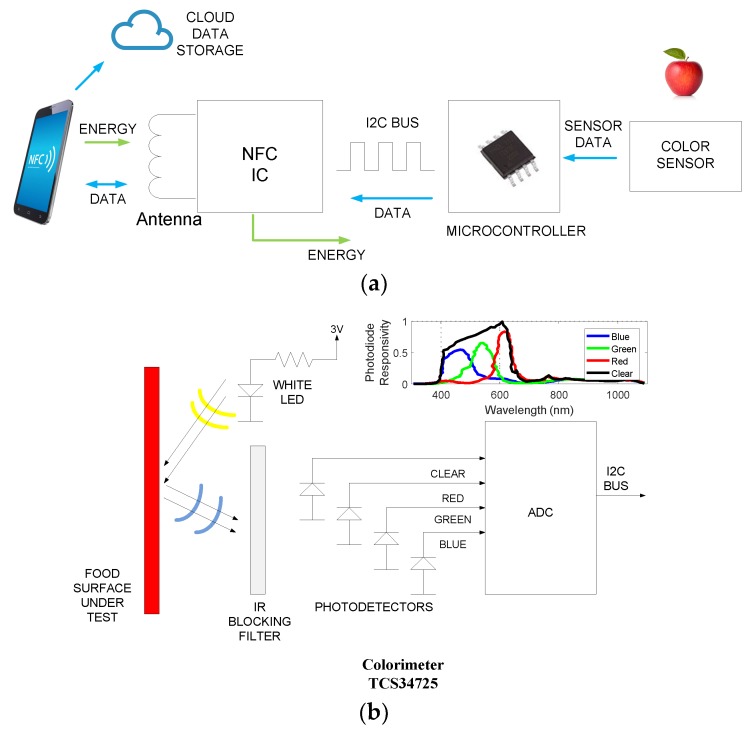
(**a**) Block diagram of the tag; (**b**) detail of the colorimeter sub-block.

**Figure 3 sensors-19-01741-f003:**
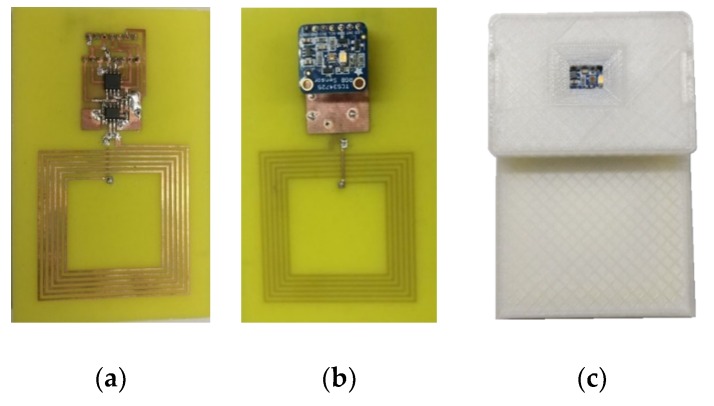
Photograph of the tag prototype: (**a**) Front side, (**b**) back side, (**c**) tag within the 3D printed enclosure.

**Figure 4 sensors-19-01741-f004:**
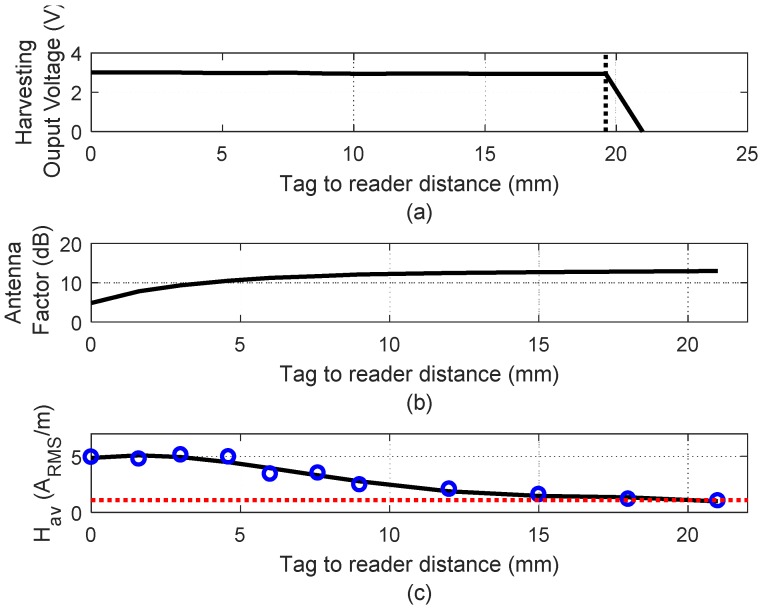
(**a**) Voltage of the harvesting NFC IC output in (V) as a function of the distance to the mobile reader; (**b**) Measured antenna factor as a function of distance to the mobile reader; (**c**) Measured magnetic field in A_RMS_/m as a function of the distance to the mobile reader.

**Figure 5 sensors-19-01741-f005:**
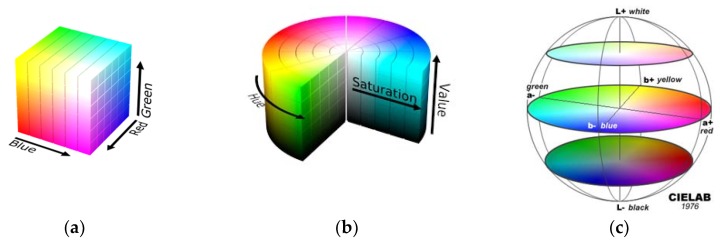
(**a**) RGB color space, (**b**) HSV color space, (**c**) CIELab color space.

**Figure 6 sensors-19-01741-f006:**
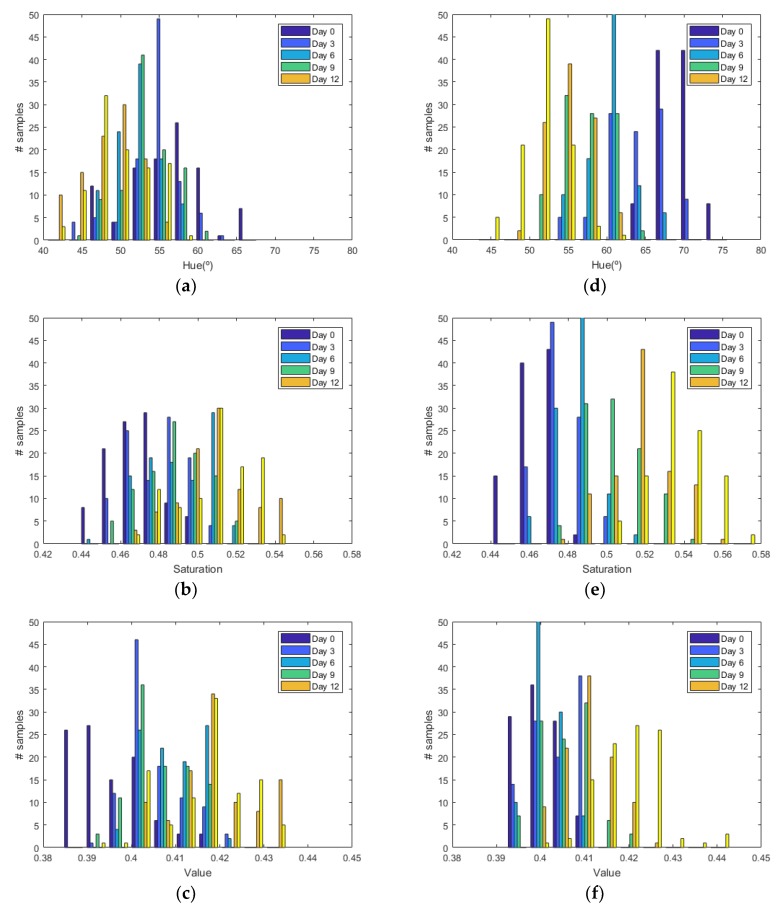
Histogram for the golden apple for different days: Histogram of the hue (**a**), saturation (**b**) and value (**c**) parameters in the fridge. Histogram of the hue (**d**), saturation (**e**) and value (**f**) parameters at room temperature.

**Figure 7 sensors-19-01741-f007:**
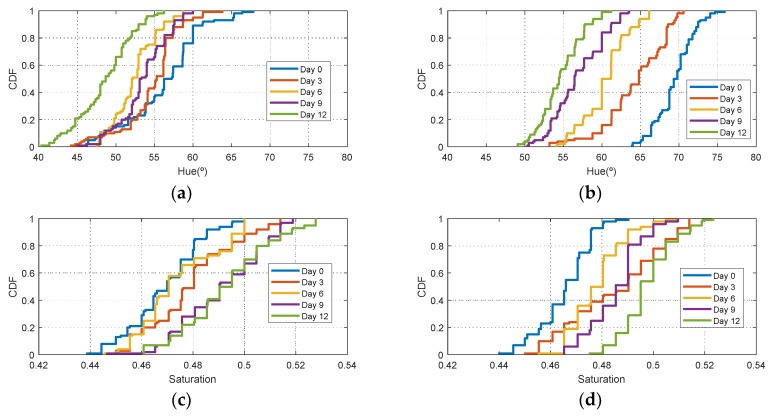
Cumulative Distribution Function (CDF) of the hue parameter for the golden apple in the fridge (**a**) and at room temperature (**b**) as a function of the number of days out of the fridge. Cumulative Distribution Function (CDF) of the saturation parameter for the golden apple in the fridge (**c**) and at room temperature (**d**) as a function of the number of days out of the fridge.

**Figure 8 sensors-19-01741-f008:**
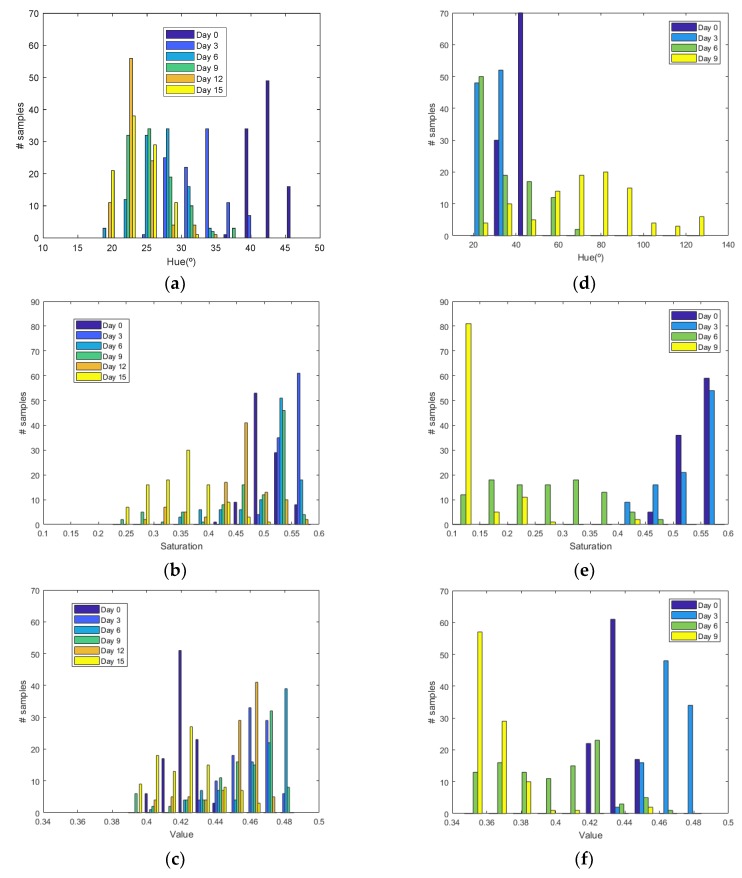
Histogram for a banana for different days: Histogram of the hue (**a**), saturation (**b**) and value (**c**) parameters in the fridge. Histogram of the hue (**d**), saturation (**e**) and value (**f**) parameters at room temperature.

**Figure 9 sensors-19-01741-f009:**
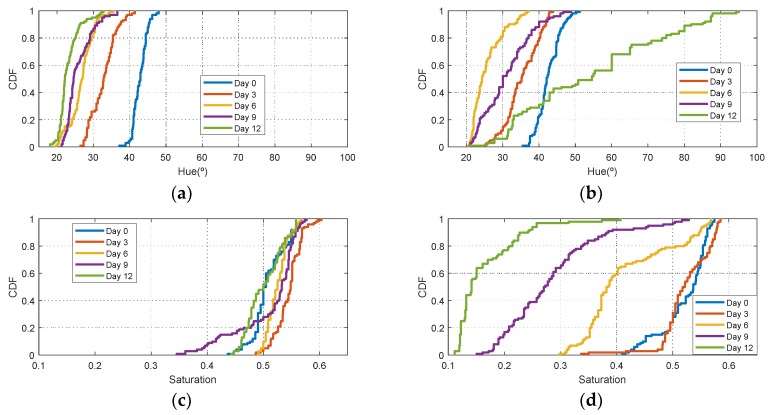
Cumulative Distribution Function (CDF) of the hue parameter for the banana in the fridge (**a**) and at room temperature (**b**) as a function of the number of days. Cumulative Distribution Function (CDF) of the saturation parameter for the banana in the fridge (**c**) and at room temperature (**d**) as a function of the number of days.

**Figure 10 sensors-19-01741-f010:**
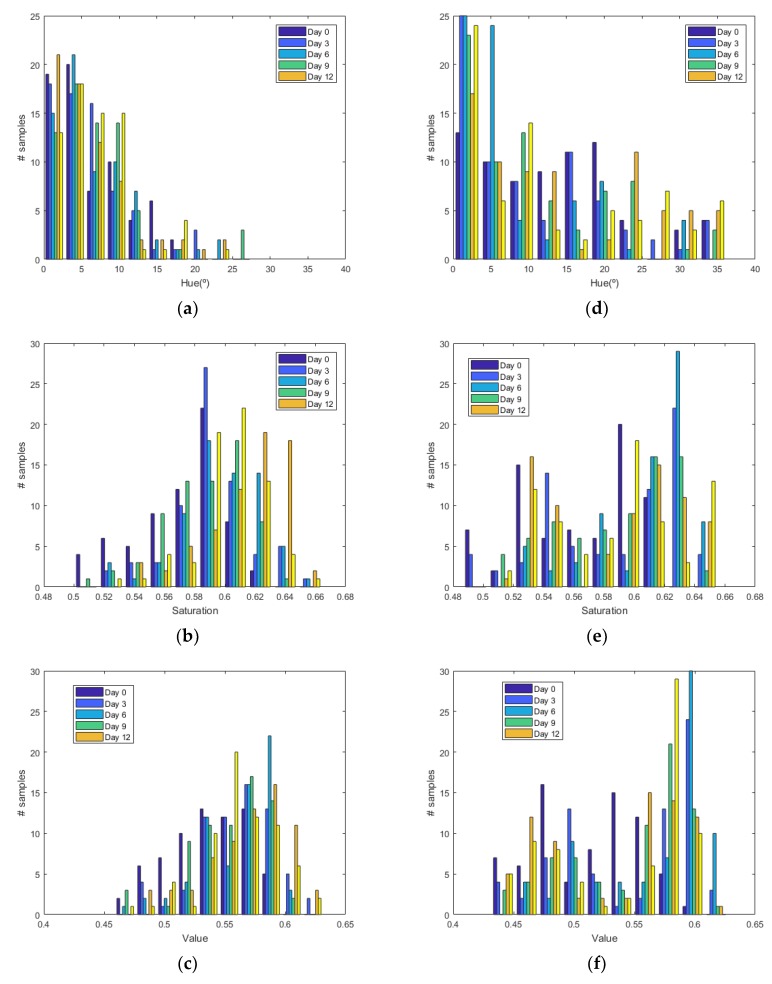
Histogram for a red apple for different days: Histogram of the hue (**a**), saturation (**b**) and value (**c**) parameters in the fridge. Histogram of the hue (**d**), saturation (**e**) and value (**f**) parameters at room temperature.

**Figure 11 sensors-19-01741-f011:**
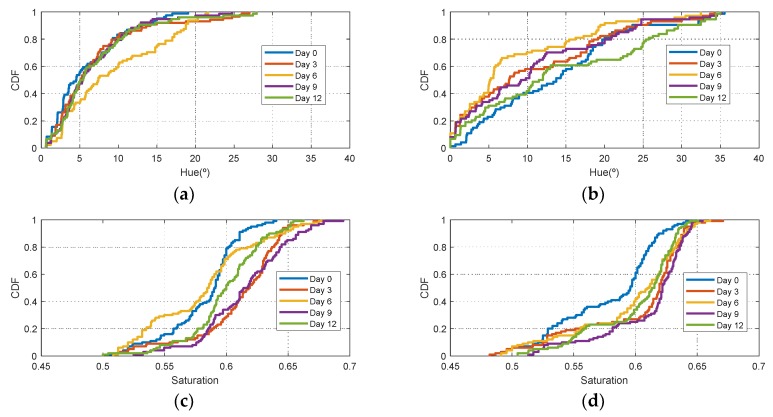
Cumulative Distribution Function (CDF) of the Hue parameter for the red apple in the fridge (**a**) and at room temperature (**b**) as a function of the number of days. Cumulative Distribution Function (CDF) of the saturation parameter for the red apple in the fridge (**c**) and at room temperature (**d**) as a function of the number of days.

**Figure 12 sensors-19-01741-f012:**
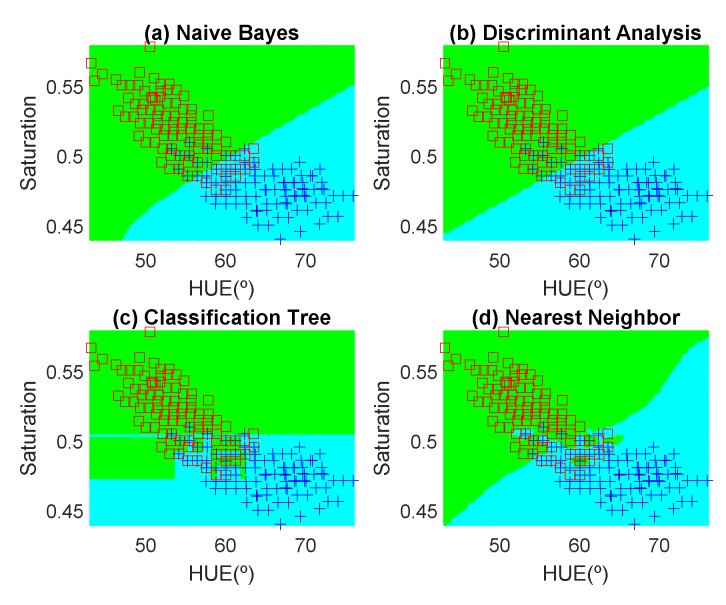
Decision boundaries and scatter plot (class 1 squares, class 2, crosses) for the golden apple. (**a**) Naive Bayes, (**b**) linear discriminant analysis, (**c**) classification tree and (**d**) Nearest Neighbor (k = 5).

**Figure 13 sensors-19-01741-f013:**
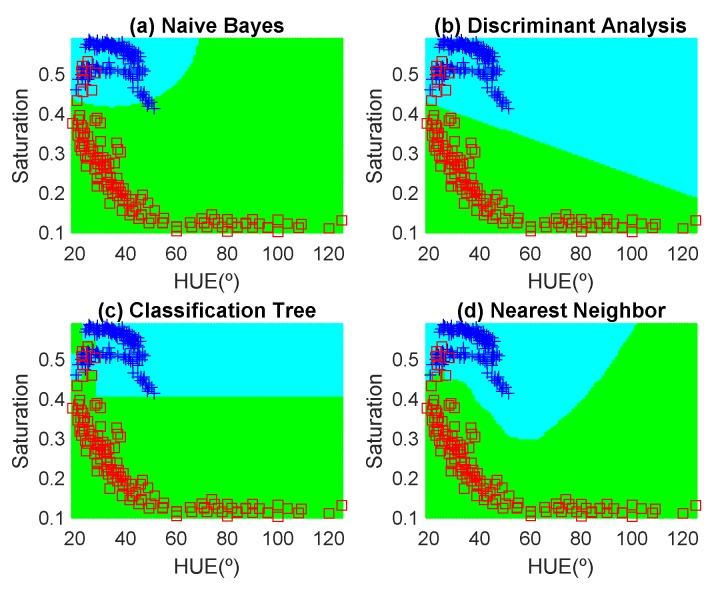
Decision boundaries and scatter plot (class 1 squares, class 2, crosses) for the banana. (**a**) Naive Bayes, (**b**) linear discriminant analysis, (**c**) classification tree and (**d**) Nearest Neighbor (k = 5).

**Figure 14 sensors-19-01741-f014:**
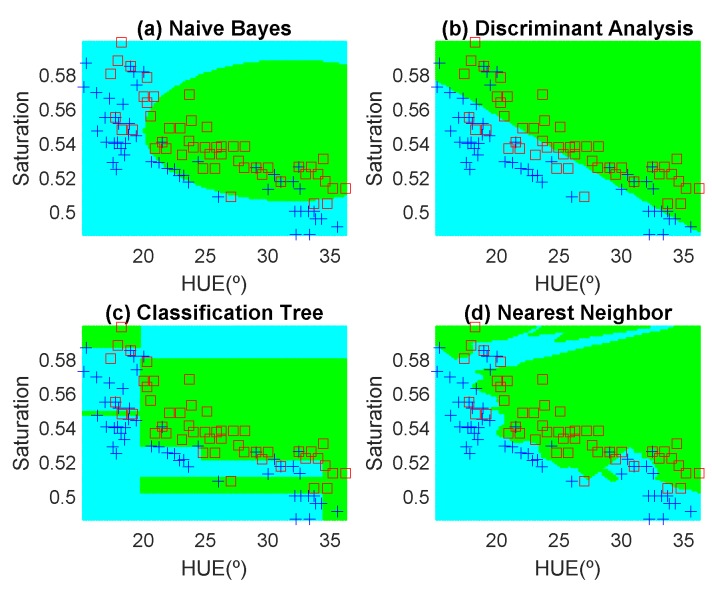
Decision boundaries and scatter plot (class 1 squares, class 2, crosses) for the red apple. (**a**) Naive Bayes, (**b**) linear discriminant analysis, (**c**) classification tree and (**d**) Nearest Neighbor (k = 5).

**Figure 15 sensors-19-01741-f015:**
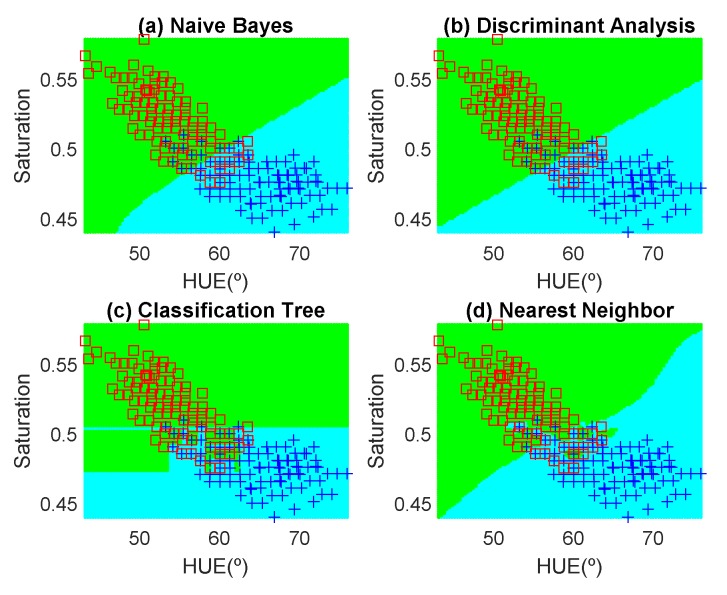
Decision boundaries and scatter plot (class 1 squares, class 2, crosses) for the golden apple. (**a**) Naive Bayes, (**b**) linear discriminant analysis, (**c**) classification tree and (**d**) Nearest Neighbor (k = 5).

**Figure 16 sensors-19-01741-f016:**
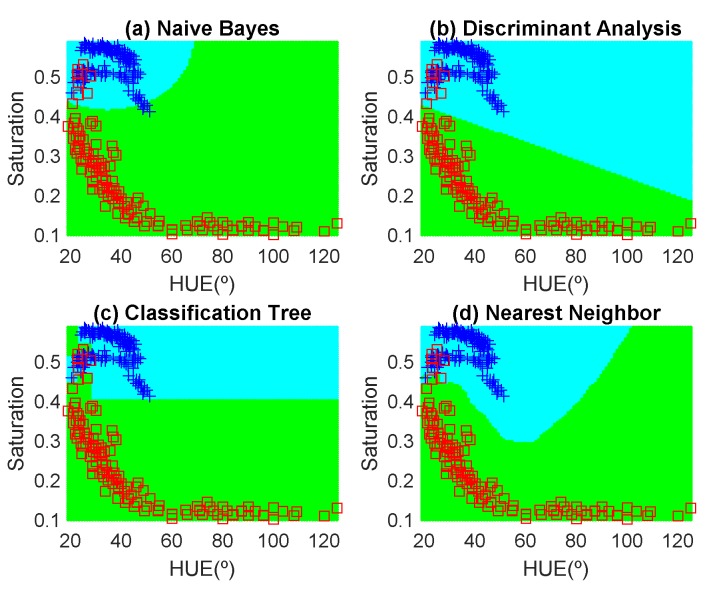
Decision boundaries and scatter plot (class 1 squares, class 2, crosses) for the banana. (**a**) Naive Bayes, (**b**) linear discriminant analysis, (**c**) classification tree and (**d**) Nearest Neighbor (k = 5).

**Figure 17 sensors-19-01741-f017:**
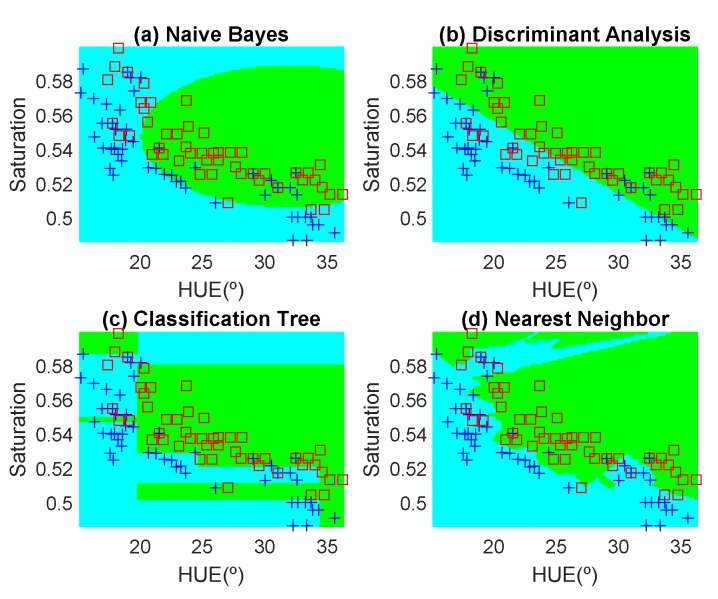
Decision boundaries and scatter plot (class 1 squares, class 2, crosses) for the red apple. (**a**) Naive Bayes, (**b**) linear discriminant analysis, (**c**) classification tree and (**d**) Nearest Neighbor (k = 5).

**Figure 18 sensors-19-01741-f018:**
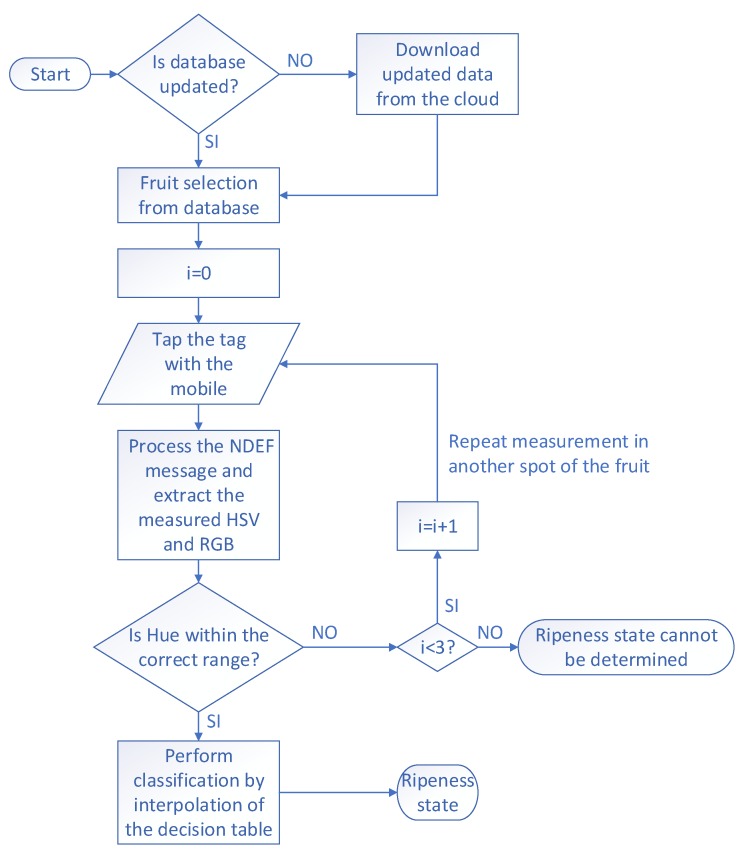
Flowchart of the mobile application.

**Figure 19 sensors-19-01741-f019:**
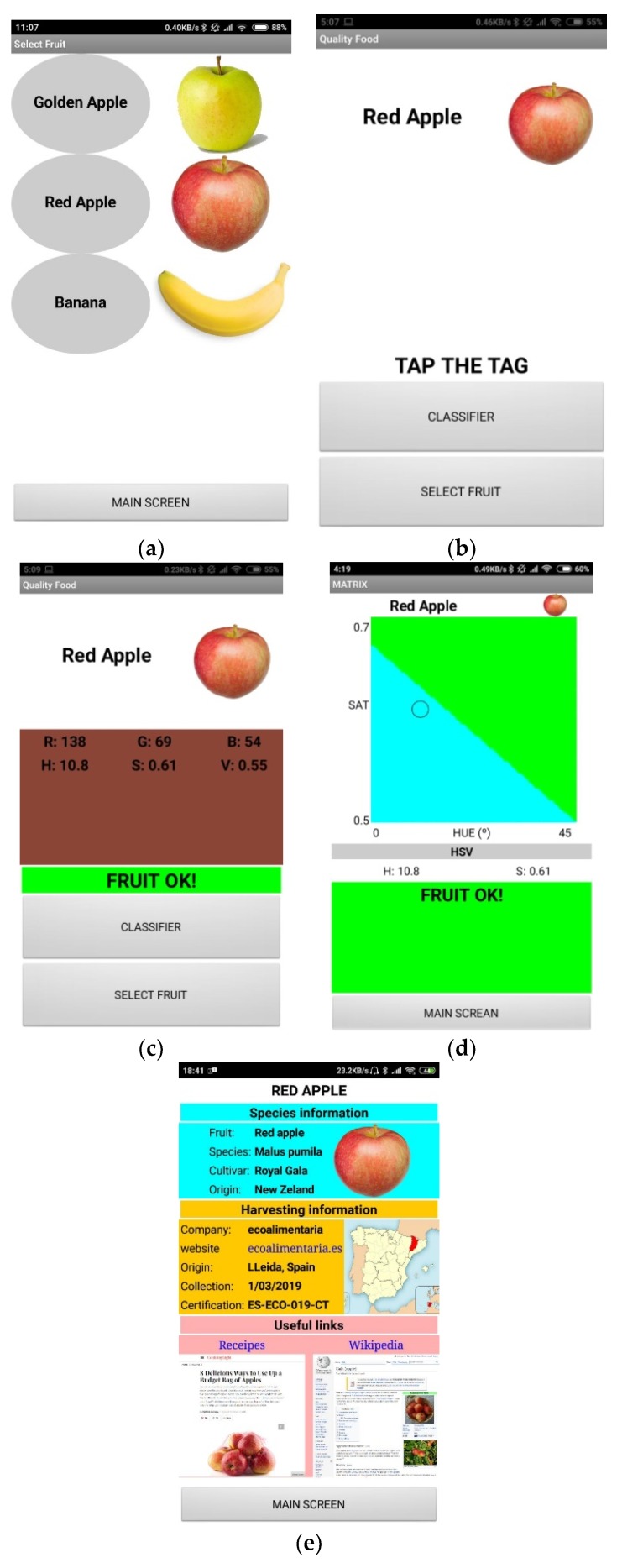
Phone screen of the developed application. (**a**) Fruit selection, (**b**) screen indicating to tap the tag, (**c**) representation of the detected color, (**d**) decision boundaries of the training, (**e**) additional user information.

**Figure 20 sensors-19-01741-f020:**
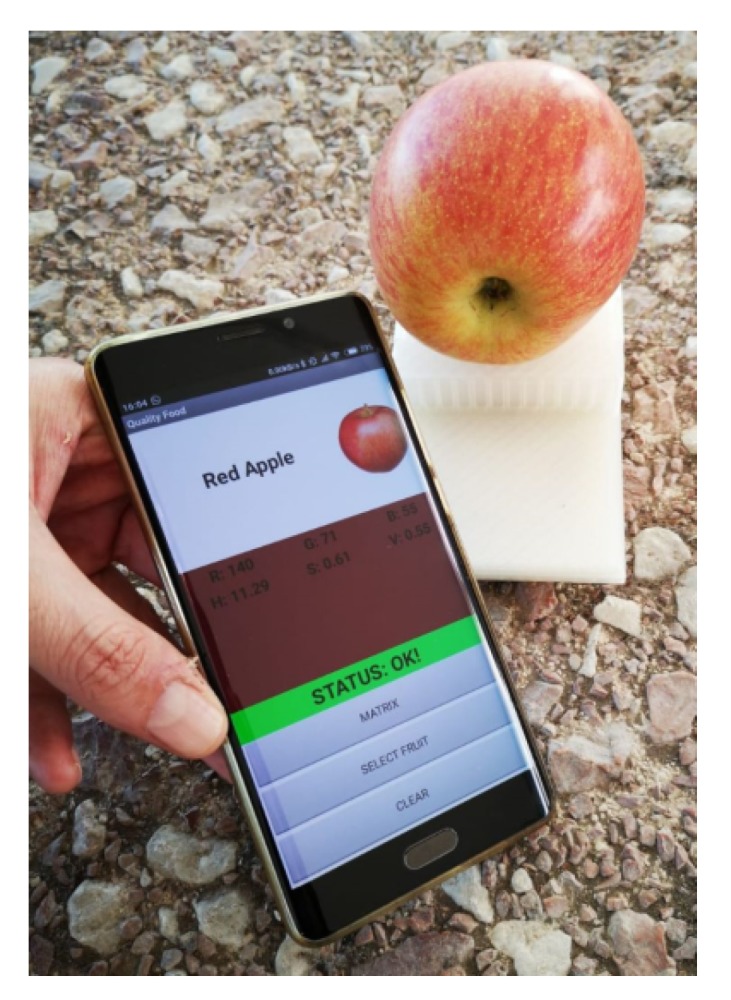
Measurement of a red apple using the designed application.

**Table 1 sensors-19-01741-t001:** Current consumption and components cost.

Component	Current Consumption (µA)	Approximate Cost for Large Quantities ($)
M24LR04E	400	0.3
TCS34725	235	1.5
LED	2000	0.03
ATTiny85	300	1.0
Case and other passive components	-	1.0
Total	2935	3.9

**Table 2 sensors-19-01741-t002:** Repeatability of the measured HSV parameters of three color samples.

Color	Sample 1 (Red)	Sample 2 (Green)	Sample 3 (Blue)
Average HUE	0.31	150.41	205.34
Normalized HUE deviation	0.09%	0.03%	0.12%
Normalized Difference HUE between sensors	0.06%	0.19%	0.46%
Average Saturation parameter	0.62	0.32	0.58
Normalized Saturation deviation	0.6%	0.14%	1.03%
Normalized Difference Saturation between sensors	1.06%	1.04%	1.00%
Average Value	0.62	0.38	0.47
Normalized Saturation deviation	0.47%	0.64%	0.45%
Normalized Difference Value between sensors	0.4%	1.0%	0.70%

**Table 3 sensors-19-01741-t003:** Average readings of HSV and L*a*b of 100 samples. Comparative change of each parameter over time.

Fruit	Days	H	S	V	L*	a*	b*
Golden Apple	Day 0	59.4	0.48	0.40	41.1	−7.3	26.6
	Day 15	40.0	0.60	0.46	40.9	3.6	30.2
	Δ%	5.4	12.00	6.00	0.1	−4.3	1.4
Red Apple	Day 0	17.4	0.58	0.51	37.9	20.6	22.1
	Day 15	23.2	0.60	0.51	39.4	17.6	25.8
	Δ%	1.6	2.00	0.00	1.5	1.2	1.4
Banana	Day 0	41.2	0.55	0.44	40.0	2.0	27.1
	Day 9	72.5	0.13	0.36	38.0	−2.4	6.0
	Δ%	8.7	42.00	8.00	2.0	−1.7	8.2

**Table 4 sensors-19-01741-t004:** Confusion matrix for ripeness grading identification for different classifiers.

Fruit	Classifier	TP %	FP %	FN %	TN %	Accu. %
Golden Apple	Naive Bayes	73.66	6.67	26.33	93.33	83.50
	Linear Discriminant Analysis	73.00	8.67	27.00	91.33	82.17
	Decision Tree	76.33	12.33	23.67	87.67	82.00
	Nearest Neighbor	81.33	13.33	18.67	86.67	84.00
	Nearest Neighbor k = 5	80.00	13.67	20.00	86.33	83.17
	SVM	68.00	2.00	32.00	98.00	83.00
Banana	Naive Bayes	90.00	4.00	10.00	96.00	93.00
	Linear Discriminant Analysis	90.50	5.00	9.50	95.00	92.75
	Decision Tree	84.50	3.50	15.50	96.50	90.50
	Nearest Neighbor	87.00	13.00	13.00	87.00	87.00
	Nearest Neighbor k = 5	87.00	3.00	13.00	97.00	92.00
	SVM	90.50	5.50	9.50	94.50	92.50
Red apple	Naive Bayes	73.33	46.67	26.67	53.33	63.33
	Linear Discriminant Analysis	86.67	40.00	13.33	60.00	73.33
	Decision Tree	63.33	40.00	36.37	60.00	61.67
	Nearest Neighbor	60.00	66.67	40.00	33.33	46.67
	Nearest Neighbor k = 5	80.00	36.67	20.00	63.33	71.67
	SVM	96.67	90.00	3.3	10.00	53.33
	QDA	86.67	26.67	13.33	73.33	80.00

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
