# Peer review of "Color Measurement and Analysis of Fruit with a Battery-Less NFC Sensor"

_sensors, 2019, doi:10.3390/s19071741_

Round 1

Reviewer 1 Report

This manuscript presents a low-cost passive system that uses NFC as energy harvesting method and a colorimeter in order to obtain an objective response of the colour presented to classify and determine the grade of ripeness in fruits. The authors have developed a smartphone customized application to show the user the result of the analysis. 

Although the idea is quite interesting, I consider that the novelty is not enough for this manuscript. Besides that fact, there are other comments that could be addressed by the authors:

- Fig. 7 is for golden apple, and a faster shift can be seen in the figure when the fruit is at room temperature. However, in Fig. 9, for bananas, the situation is only visible for day 9. Therefore, the sentence in Page 6, line 192 “It can be observed that the color variation is smooth in the case of the fruit conserved in the fridge, whereas the shift is faster at room temperature” is not always true.

- In page 2, the authors state “The application in the smartphone reads the message, classifies the quality of the food and gives additional information about the origin or some other useful information.” However, no other information is presented in the application.

- In general, the x axis scale in not the same, therefore is not possible to compare figures and values of CDF and HSV for the different cases that are shown.

- Is any information of temperature, humidity or days after harvesting stored in the system? There is no coincidence in values of HSV depending on those factors?

-  What is the value for the error in the system during the acquisition? Since authors state that “Hue changes 1.6% for red apples, 5.4% for golden apples and 8.7% for bananas; whereas saturation (S) changes 2% for red apples, 12% for golden apples 226 and 42% for bananas.”. Is the error in the system less than 1.6% in the measurement of H? How many fruits did you analyse? I cannot see if there are errors bars in any graphic, and there is no any information about the error of the system in the calibration procedure.

- Fruits have many colors: a banana have yellow areas but also brown and black. Red apples have different red and yellow areas and golden apple have small brown dots sometimes. Although the authors pointed out how they filtered measurements, are all the possibilities included in the software? Besides, sometimes fruit is ripened just in one side, being good in the other part. I don’t really see the point of this application.

-   Why is the case white, that reflect the color instead of black?

Author Response

Responses:

We understand that the citation of the previous works is fundamental. Thus, we have improved the citation of the text. I have included a recent published work by the authors (accepted during the review process that now is available) where the hardware implementation of the NFC and colorimeter sensor is explained in detail. In this work, only a summary has been presented to improve the readability and comprehension. However, this work is focused in another application; the firmware, the mobile software and the processing (with high weight in this paper) in are completely different and novel, and in fact, these aspects are the ones mostly addressed in the paper. In addition, some paragraphs, already correctly referenced in the original paper, are rewritten with other words and cited again.

M. Boada, A. Lazaro, R. Villarino and D. Girbau, "Battery-Less NFC Sensor for pH Monitoring," in IEEE Access, vol. 7, pp. 33226-33239, 2019. doi: 10.1109/ACCESS.2019.2904109

Reviewer 1:

Comments: This manuscript presents a low-cost passive system that uses NFC as energy harvesting method and a colorimeter in order to obtain an objective response of the colour presented to classify and determine the grade of ripeness in fruits. The authors have developed a smartphone customized application to show the user the result of the analysis.

Although the idea is quite interesting, I consider that the novelty is not enough for this manuscript.

Answer:We agree with the reviewer that the classification of the state of fruit from the color is not novel (see for example the references included in the introduction). But the main novelty of the work is the procedure of classification using a low-cost battery-less NFC device with friendly utilization by the final user without previous knowledge about classification techniques. In the following the authors answer the constructive comments that have help us to improve the work.

Comments: Besides that fact, there are other comments that could be addressed by the authors:

Comments: Fig. 7 is for golden apple, and a faster shift can be seen in the figure when the fruit is at room temperature. However, in Fig. 9, for bananas, the situation is only visible for day 9. Therefore, the sentence in Page 6, line 192 “It can be observed that the color variation is smooth in the case of the fruit conserved in the fridge, whereas the shift is faster at room temperature” is not always true.

Answer: We agree to the reviewer that the case of banana is a bit particular. In the case of the banana, the variation is appreciated better in the other parameters, for example in the saturation. The CDF for the saturation parameters have been added for the three fruits.  A mistake in the legend of the figure 7 has also been corrected. The phase has been rewritten:

“In Figure 7 the Cumulative Distribution Function (CDF) of the hue and saturation parameters for the golden apple in the fridge and at room temperature as a function of the number of days is shown. It can be observed that the color variation is smoother for the fruit conserved in the fridge than at room temperature. Therefore, the ripeness grade is also a function of the environment parameters. Figures 8-11 show the histograms of HSV measurements and CDF of the hue parameter for the banana (Figures 8-9) and the red apple (Figures 10-11). Similar behavior is obtained for the red apple case. In the case of banana, the fruit is degraded very fast for more than 9 days at room temperature. This fact can be appreciated better in the variation of the saturation value for the banana at room temperature. As a conclusion, the combination of HUE and saturation value can be used to study the degree of ripeness.”

Comments: In page 2, the authors state “The application in the smartphone reads the message, classifies the quality of the food and gives additional information about the origin or some other useful information.” However, no other information is presented in the application.

Answer: The work is focused on the classification, however, when the app is connected to the database to update the classification, data can be also download with custom information that can be useful for the user as additional information that you comment or useful web links. This information is included in the app in another screen that is open when the user pushes over the image of the fruit. This screen is added in Fig.19(e). A brief comment has been added in the text after the figure 19:

“When the user push over the image of the fruit a new screen is open (Figure 19e) where the user can consult additional information about the product such as the origin, date of collection, web links, etc.

Comments:In general, the x axis scale in not the same, therefore is not possible to compare figures and values of CDF and HSV for the different cases that are shown.

Answer:Thank you for the comment. The x axis scales have been changed in order to improve the comparison of the different cases.

Comments:Is any information of temperature, humidity or days after harvesting stored in the system? There is no coincidence in values of HSV depending on those factors?

Answer: The grade of ripeness depends of the humidity, temperature or days after harvesting but we have not used this information in the classifier. This influence is highlighted in the introduction:

" The surface color of the food is the first element that the consumer observes and has a great influence on the consumer’s selection [1]. It depends on various factors, including the temperature, humidity, and biochemical changes that occur during growth, maturation, postharvest handling and processing [1]. In consequence, food color is an excellent indicator of its quality because it takes into account these parameters and it is one of the most broadly measured product quality attributes in postharvest handling and in food processing research and industry [1]."

This dependence is shown in the comparison with the fruit inside and outside the fridge. For the same days, the difference in the temperature produces different degrees of ripeness. The color (HSV) as a measure of ripeness integrates this information. In a real application, these parameters are often not controlled or unknown or depend on the place where fruits are located. Therefore, the temperature or humidity cannot be used as features for the classifier. To this end, a datalogger should be used which increases the cost of the system. The instantaneous information of temperature or humidity can be also measured using a battery-less NFC tag. In a recent work of the authors a soil moisture sensor with ambient temperature and humidity has been presented in [21]. However, it would need all the story and a battery to save the data. Here the aim is only use the color as a memory.

Comments: What is the value for the error in the system during the acquisition? Since authors state that “Hue changes 1.6% for red apples, 5.4% for golden apples and 8.7% for bananas; whereas saturation (S) changes 2% for red apples, 12% for golden apples 226 and 42% for bananas.”. Is the error in the system less than 1.6% in the measurement of H? How many fruits did you analyse? I cannot see if there are errors bars in any graphic, and there is no any information about the error of the system in the calibration procedure.

Answer: Measurements with a colorimeter have higher precision compared to measurements done with cameras (smartphone) because the illumination is more controlled (see for example comparison in ref.[21]). Some measurements have been included to show the accuracy of the acquisition system and determine the difference between measurements during calibration and after. Three samples with different colors (one red, one green and another blue) have been measured with 3 colorimeters, 200 times for each color and colorimeter. A summary has been included in Table 2. This table shows the average (mean) value of each HSV sample, the normalized standard deviation with respect to the maximum range of the parameter, and the normalized maximum difference between the colorimeters. It can be observed that the differences and the deviation is smaller than the typical difference observed with the days. The small deviation is the measurement of the resolution of the system. It can be observed in the histograms that the distribution is flat, therefore, the noise is due to the discretization noise of the internal AD in the colorimeter IC. Two new references where the accuracy of color measurements between colorimeters and spectrometers is compared have been included (ref. [29] and [30]). The following text has been included to explain the new table with the results.

“Colorimeters have been used in other applications with good accuracy compared to other color measurement systems such as spectrometers [29][30]. However, it is needed to check the repeatability and accuracy of the measurements. It has been investigated through the measurement of three samples with different colors. Each sample has been measured 200 times with three colorimeters using the same IC model. Table 2 summarizes the main results. This table shows the average values of each HSV sample, the normalized standard deviation with respect to the maximum range of the parameter, and the normalized maximum difference between the colorimeters. It can be observed that the differences and the deviation is smaller than the typical difference observed with the days. The error is uniformly distributed, showing that the main source of error is discretization noise due to the internal analog to digital conversion in the colorimeter IC. The difference in the HSV values between different ICs are small (typically under 1%). It is assumed that the calibration and the measurement by the final user is done with the same model of colorimeter to improve the repeatability. However, small differences can be found between colorimeters and spectrometers [30].

We have followed the standard procedure used in SVM classifiers. In the preliminary results to test the algorithms, we have analyzed 12 fruits with 100 samples each day in different random points over the surface. The training dataset is composed by 12 fruits and samples of another fruit (not used in the training dataset) are used for testing after the training.  More details about the procedure have been included in the text:

" In order to avoid introducing any bias in the classification results, the dataset is composed of the same number of measurements each day (100 measurements). These samples have been obtained in random positions of the surface of the fruits to take into account their heterogeneous composition. In the preliminary results, 12 fruits per day are measured. Another fruit that is not included in the training dataset is used for testing."

Comments: Fruits have many colors: a banana have yellow areas but also brown and black. Red apples have different red and yellow areas and golden apple have small brown dots sometimes. Although the authors pointed out how they filtered measurements, are all the possibilities included in the software? Besides, sometimes fruit is ripened just in one side, being good in the other part. I don’t really see the point of this application.

Answer: Other source of error is the nonuniformity in the color of the surface of the fruits as it commented by the reviewer. These variations can be shown in the histograms and CDF plots and this random behavior results on a difficulty to define simple decision regions. Therefore, the objective of the work is to investigate the robustness of the different classifiers with three types of fruits. The golden apples have small pigments which are often uniformly distributed. Due to its small size, normally they are averaged with the surrounding points within the area detected by the colorimeter and their influence is small. The red apples used have a gradual variation over the surface. The bananas have different areas and when they degrade the color changes a lot.  The area analyzed by the colorimeter is small compared to the area that can be analyzed with other methods such as a camera where can be used some segmentation techniques to mitigate the problem. In this sense, the use of camera has advantages but the repeatability is worst due to the illumination conditions. Therefore, the main source of error is the local point considered in each measurement that have the same color due to defects or simply because it is not a homogenous color or have pigments as comments the reviewer. In order to solve this drawback, if the measured color falls outside the analyzed colorspace range during the training, the mobile app shows a warning dialog to the user, who can select other points in the surface to try to avoid these regions. The bandpass range of colors that is considered in the training is chosen from the histograms and the density of points in the scatter plot of all the points as a function of the Hue and Saturation value. Typically defects fall in a different range of colors, can be filtered if its probability is very high and can reduce the classifier accuracy. The case of red apple is the worst case because it has areas where the color gradually changes from yellow/green to red. In this case it is only considered the red color range (H<50º). In the training, the space of the features can not be continuous, however these situations can introduce errors in simplest classifier such as LDA where the boundary decision is a line. This filter is explained in the flowchart. However, a brief comment is included in the text when it is commented the flowchart:

" The application sends a warning to the user when the point is outside the decision boundaries, because the analyzed point falls on some area with high concentration of pigments or there is a defect on the surface with different color. After that, the user can select another point (see the flowchart in Figure 18)."

A brief discussion of this drawback has been inserted in the conclusions:

“A drawback of the presented colorimeter is the small size of the analyzed area. If large defects or pigments with different color fall in this area, the points are treated as outliers and the measurement must be repeated. The portable device and algorithm described in this work could be extended to another applications, such as obtaining the grade of ripeness of the fruit during the harvesting, whenever the classification algorithms were trained with samples of a different grade of maturation.

Comments:  Why is the case white, that reflect the color instead of black?

Answer: We have checked that the color of the case does not affect the measurement because the colorimeter has a window and the fruits cover all the window, avoiding the penetration of external light that can modify significantly the measurement as it observed in the new repeatability measurements added. The thickness of the case is enough to be considered opaque and the size is enough to neglect important reflections in its surface. An internal white led is used to perform the measurements under controlled illumination conditions to ensure the repeatability of the measurements. The color of case can be changed. Black would be better if the samples were very small as in the case of PH sensor proposed in [21]. In the prototype, we have used PLA with white color for the 3D print but it can be easily changed. A brief comment has been included in the text:

“…A protection envelope was designed using a 3D printer. A protection envelope has been designed using a 3D printer. This case can be customized and has a window over the color sensor that is covered by the fruit. The box is opaque and it is only illuminated by the internal white led. This fact improves the accuracy and repeatability of the measurements.

Reviewer 2 Report

Interesting paper, NFC part can be further improved, mainly by optimizing the available current (high Q and tag tuning to higher frequency, by using energy storage device and optimize measurement time, and data logging using battery supported semi-passive tag, …) and – as you are doing - by using low data rate (high Q and low bandwidth).

Further on, classifying algorithms should be used for determining the ripeness of fruit for harvesting.

Please check and correct the following:                   

1. Table 2 is incorrect (bananas/red apple should be corrected in table and correct data in the statement accordingly:

»However, the variations are smaller using CIELab color space, with differences between 0.1% and 2% for L, 1.2% and 4.3% for parameter a* and 1.4% and 8.2 for parameter b*. 228«

2. Pls check again all figures and appropriate text to figures.

Author Response

Editor comments:

We understand that the citation of the previous works is fundamental. Thus, we have improved the citation of the text. I have included a recent published work by the authors (accepted during the review process that now is available) where the hardware implementation of the NFC and colorimeter sensor is explained in detail. In this work, only a summary has been presented to improve the readability and comprehension. However, this work is focused in another application; the firmware, the mobile software and the processing (with high weight in this paper) in are completely different and novel, and in fact, these aspects are the ones mostly addressed in the paper. In addition, some paragraphs, already correctly referenced in the original paper, are rewritten with other words and cited again.

M. Boada, A. Lazaro, R. Villarino and D. Girbau, "Battery-Less NFC Sensor for pH Monitoring," in IEEE Access, vol. 7, pp. 33226-33239, 2019. doi: 10.1109/ACCESS.2019.2904109

Reviewer 2:

We would like to thank the editor and the reviewers for the helpful suggestions and appropriate comments which contributed to improve the quality of the manuscript. We have modified the paper following the comments or remarks, answering their questions. We believe we have solved all the concerns and we include hereafter a list of clarifications.

Sincerely,

The authors

Comments: Interesting paper, NFC part can be further improved, mainly by optimizing the available current (high Q and tag tuning to higher frequency, by using energy storage device and optimize measurement time, and data logging using battery supported semi-passive tag, …) and – as you are doing - by using low data rate (high Q and low bandwidth).

Answer: Details about the hardware implementation have been described in detail in other recent publication of the authors published during the revision of this work. This reference has been included in the manuscript (see ref.21). The aim of this work is to show the possibility of using a battery-less device for colorimetric applications. The same NFC IC can be powered with an external battery, then the door is open to other applications such as data logging. In this case, the communication by NFC would be only used to download the data saved in the internal memory of the NFC IC or in an external memory or internal memory of the microcontroller. For battery-supported devices other communication technologies can be used such Bluetooth low energy (BLE) to download the data with higher read range. In addition, controlling the duty cycle can enlarge the life time of the batteries and sensors with higher current consumption. However, battery-assisted devices have and important drawback, that is the battery and devices with BLE have higher cost compared with battery-less option besides the problem of contamination associated to batteries. In addition, batteries can be degraded with the time or humidity and they can be toxic, and usually are forbidden to be in contact with food. A discussion and comparison of NFC-based sensors with other technologies has been done by the authors in [20]. The possibility of using high Q coils and lower data rate (bandwidth) may be a method to increase the read range however the data rates are determined by the NFC standards that are designed for fast transfer of high quantities of data saved using the NDEF message format. In our case, the length of the payload is very small. But these changes suppose the use of non-standard readers and one of the advantage of the work is the possibility of reading the data from smartphones with NFC. The read range obtained is enough to successfully read the data and feed the sensor.

A brief discussion about the cost of the prototype has been added in the text:

“The battery-less system presented is based on low-cost commercial integrated circuits. Due to the wide diffusion of NFC systems, the cost of NFC IC and the low-power microcontroller are smaller than 1 $. The price of the colorimeter IC is around 1.5 $. Thus the overall cost of the tag including the envelope can be under 5 $ considering large volumes of production (see estimation in Table 1). This cost is noticeable lower than professional colorimeters or spectrometers that are typically starting from 600-1000 $. In addition, the presented system is easy to use and is highly customizable depending on the final application. The lack of battery is another advantage because it avoids the need to replace or recharge the batteries whereas, enlarging the durability of the devices and avoiding the component with higher cost. In addition, the batteries contain toxic components that can contaminate the food in addition to generate non-recyclable waste.”

 “The calibration was done by a different colorimeter (same model but a different one) but connected directly to the computer, through the programing connector. The data is transferred to the database to perform the training of the classifiers. The samples are taken in different positions to consider the variations of color on the surface. It is important to note that the samples have been kept considering typical conditions (typical range of temperatures and humidity) that the end user will find. To ensure these steps it is preferable that the calibration is carried out by the manufacturer in a qualified laboratory.”

“In order to avoid introducing any bias in the classification results, the dataset is composed of the same number of measurements each day (100 measurements). These samples have been obtained in random positions of the surface of the fruits to take into account their heterogeneous composition. In the preliminary results, 12 fruits per day are measured. Another fruit that is not included in the training dataset is used for testing.

“The application sends a warning to the user when the point is outside the decision boundaries, because the analyzed point falls on some area with high concentration of pigments or there is a defect on the surface with different color. After that, the user can select another point (see the flowchart in Figure 18).”

 A discussion about the effect of the quality factor in the read range has been briefly added at the end of section 2:

“Theoretically, as the data rate between sensor and reader is not excessively high, the bandwidth could be reduced. Consequently, coils with high-quality factors to increase energy transfer efficiency can be designed. However, the loaded quality factor of the tag is low [20][21] and it is determined by the low IC equivalent resistance. Nevertheless, this low-Q factor has an advantage because the system is more robust to the detuning due to the proximity of metals (metallic case of the modern smartphones) or high permittivity materials (for example the body).

Comments: Further on, classifying algorithms should be used for determining the ripeness of fruit for harvesting.

Answer: The application suggested for the reviewer is very interesting and it has been proposed in some reference cited in the work [3]. The proposed technique can be used for determining the ripeness if the classifying algorithms are trained with different samples with different degree of ripeness. It is expected that the change in the color will be higher than the one considered here and the techniques can work.  The use of a portable and low cost device can help to perform this process. This potential application has been highlighted in the conclusions as a future research line:

“The portable device and algorithm described in this work could be extended to another applications, such as obtaining the grade of ripeness of the fruit during the harvesting, whenever the classification algorithms were trained with samples of a different grade of maturation.

Comments:

Please check and correct the following:                  

1. Table 2 is incorrect (bananas/red apple should be corrected in table and correct data in the statement accordingly:

»However, the variations are smaller using CIELab color space, with differences between 0.1% and 2% for L, 1.2% and 4.3% for parameter a* and 1.4% and 8.2 for parameter b*. 228«

2. Pls check again all figures and appropriate text to figures.

Answer: Thank you again. The mistake in table 2 has been corrected (the label bananas and red apple must be interchanged).

Following your suggestions, the figures are checked. In addition, the axis of histograms has been changed to improve the comparison between the different cases.

Reviewer 3

Comments: The paper is well structured and (very) well written. The description of the system is clear and concise. The training process is carried out with Matlab; this might be the most difficult step in order to bring this system to a wide implementation. It is mentioned that this can be performed in a laboratory or by the manufacturer. Thus, it is evident that the surrounding conditions will play an important role in this process. Therefore, it would be desirable to add a paragraph or two giving additional details on how the training can be made in particular environments. Finally, the overall cost of the system could be compared to an approximate of alternatives that have been used for the same purpose in order to highlight the attractiveness of the proposed system

Answer: Thank you for your comments. A brief discussion about the cost and alternative are included in the text:

“The battery-less system presented is based on low-cost commercial integrated circuits. Due to the wide diffusion of NFC systems, the cost of NFC IC and the low-power microcontroller are smaller than 1 $. The price of the colorimeter IC is around 1.5 $. Thus the overall cost of the tag including the envelope can be under 5 $ considering large volumes of production (see estimation in Table 1). This cost is noticeable lower than professional colorimeters or spectrometers that are typically starting from 600-1000 $. In addition, the presented system is easy to use and is highly customizable depending on the final application. The lack of battery is another advantage because it avoids the need to replace or recharge the batteries whereas, enlarging the durability of the devices and avoiding the component with higher cost. In addition, the batteries contain toxic components that can contaminate the food in addition to generate non-recyclable waste.

In addition, some comments have been included about the procedure for training. New measurements about the repeatability and reproducibility of the measurement have been included. Also, some comments about the filtering procedure to filter outliers have been inserted.

“The calibration was done by a different colorimeter (same model but a different one) but connected directly to the computer, through the programing connector. The data is transferred to the database to perform the training of the classifiers. The samples are taken in different positions to consider the variations of color on the surface. It is important to note that the samples have been kept considering typical conditions (typical range of temperatures and humidity) that the end user will find. To ensure these steps it is preferable that the calibration is carried out by the manufacturer in a qualified laboratory.

Reviewer 3 Report

The authors of this paper present the design of a classification system to assess fruit ripeness, which is based on sensing the color with an NFC tag. In order to do so, a color sensor and a microcontroller connected to an NFC chip are used. Then, a smartphone is used to read and activate the NFC tag. The aim is to classify the state (quality) of fruits as the function of the number of days outside the fridge according to the color information. The authors provide a detailed description of each part of the system built. The choice of the color space is particularly interesting given the conditions under which the proposed system operates. The statistics measured provide a very good indicator for the ripeness of fruit since this allows the authors to decide that hue angle and saturation are well suited choices for their system.

The paper is well structured and (very) well written. The description of the system is clear and concise. The training process is carried out with Matlab; this might be the most difficult step in order to bring this system to a wide implementation. It is mentioned that this can be performed in a laboratory or by the manufacturer. Thus, it is evident that the surrounding conditions will play an important role in this process. Therefore, it would be desirable to add a paragraph or two giving additional details on how the training can be made in particular environments. Finally, the overall cost of the system could be compared to an approximate of alternatives that have been used for the same purpose in order to highlight the attractiveness of the proposed system.

Author Response

Editor comments:

We understand that the citation of the previous works is fundamental. Thus, we have improved the citation of the text. I have included a recent published work by the authors (accepted during the review process that now is available) where the hardware implementation of the NFC and colorimeter sensor is explained in detail. In this work, only a summary has been presented to improve the readability and comprehension. However, this work is focused in another application; the firmware, the mobile software and the processing (with high weight in this paper) in are completely different and novel, and in fact, these aspects are the ones mostly addressed in the paper. In addition, some paragraphs, already correctly referenced in the original paper, are rewritten with other words and cited again.

M. Boada, A. Lazaro, R. Villarino and D. Girbau, "Battery-Less NFC Sensor for pH Monitoring," in IEEE Access, vol. 7, pp. 33226-33239, 2019. doi: 10.1109/ACCESS.2019.2904109

Reviewer 3:

We would like to thank the editor and the reviewers for the helpful suggestions and appropriate comments which contributed to improve the quality of the manuscript. We have modified the paper following the comments or remarks, answering their questions. We believe we have solved all the concerns and we include hereafter a list of clarifications.

Sincerely,

The authors

Comments:

The paper is well structured and (very) well written. The description of the system is clear and concise. The training process is carried out with Matlab; this might be the most difficult step in order to bring this system to a wide implementation. It is mentioned that this can be performed in a laboratory or by the manufacturer. Thus, it is evident that the surrounding conditions will play an important role in this process. Therefore, it would be desirable to add a paragraph or two giving additional details on how the training can be made in particular environments. Finally, the overall cost of the system could be compared to an approximate of alternatives that have been used for the same purpose in order to highlight the attractiveness of the proposed system

Answer:

Thank you for your comments. A brief discussion about the cost and alternative are included in the text:

“The battery-less system presented is based on low-cost commercial integrated circuits. Due to the wide diffusion of NFC systems, the cost of NFC IC and the low-power microcontroller are smaller than 1 $. The price of the colorimeter IC is around 1.5 $. Thus the overall cost of the tag including the envelope can be under 5 $ considering large volumes of production (see estimation in Table 1). This cost is noticeable lower than professional colorimeters or spectrometers that are typically starting from 600-1000 $. In addition, the presented system is easy to use and is highly customizable depending on the final application. The lack of battery is another advantage because it avoids the need to replace or recharge the batteries whereas, enlarging the durability of the devices and avoiding the component with higher cost. In addition, the batteries contain toxic components that can contaminate the food in addition to generate non-recyclable waste.

In addition, some comments have been included about the procedure for training. New measurements about the repeatability and reproducibility of the measurement have been included. Also, some comments about the filtering procedure to filter outliers have been inserted.

“The calibration was done by a different colorimeter (same model but a different one) but connected directly to the computer, through the programing connector. The data is transferred to the database to perform the training of the classifiers. The samples are taken in different positions to consider the variations of color on the surface. It is important to note that the samples have been kept considering typical conditions (typical range of temperatures and humidity) that the end user will find. To ensure these steps it is preferable that the calibration is carried out by the manufacturer in a qualified laboratory.

Round 2

Reviewer 1 Report

No comments.